# Lorentz Direct Concatenation for Stable Training in Hyperbolic Neural Networks

**Eric Qu**    ZHONGHANG.QU@DUKE.EDU   and   **Dongmian Zou**    DONGMIAN.ZOU@DUKE.EDU
*Division of Natural and Applied Sciences, Duke Kunshan University, Jiangsu 215316, China*

**Editors:** Sophia Sanborn, Christian Shewmake, Simone Azeglio, Arianna Di Bernardo, Nina Miolane

## Abstract

Hyperbolic neural networks have achieved considerable success in extracting representation from hierarchical or tree-like data. However, they are known to suffer from numerical instability, which makes it difficult to build hyperbolic neural networks with deep hyperbolic layers, no matter whether the Poincaré or Lorentz coordinate system is used. In this note, we study the crucial operation of concatenating hyperbolic representations. We propose the Lorentz direct concatenation and illustrate that it is much more stable than concatenating in the tangent space. We provide some insights and show superiority of performing direct concatenation in real tasks.

**Keywords:** hyperbolic neural networks, concatenation, numerical stability

## 1. Introduction

Recent development of geometric deep learning has witnessed a boost of neural network models specifically designed for data lying in non-Euclidean domains. For data that show a hierarchical or tree-like structure, hyperbolic neural networks have been effective and successfully applied to, for instance, product recommendation (Wang et al., 2019), drug discovery (Wu et al., 2021), and action recognition (Peng et al., 2020).

One possible approach to building hyperbolic neural networks is performing neural operations in the tangent space. The earliest such model was proposed by Ganea et al. (2018), which got refined recently by Shimizu et al. (2021). If the dataset has a graph structure, it is also possible to combine hyperbolic operations in the tangent space with message passing layers in graph convolution (Chami et al., 2019; Liu et al., 2019; Bachmann et al., 2020). Although tangent spaces are effective ways to approximate the hyperbolic domain, applying exponential and logarithmic operations usually causes numerical instability, especially within the Poincaré ball model (Nickel and Kiela, 2018; Chami et al., 2019). Recently, Chen et al. (2021) proposed to use fully hyperbolic layers, which avoid going back and forth to the tangent spaces. However, their focus was on linear layers. More complex neural networks usually include different types of layers which, even combined with fully hyperbolic linear layers, may suffer from numerical instability. In this note, we propose a concatenation layer within the Lorentz model and illustrate its numerical stability. Moreover, we analyze the stability and use it to achieve a hyperbolic generative network competitive for real tasks.

We provide some preliminaries of hyperbolic geometry in Appendix A.

## 2. Motivation and Definition of Lorentz Direct Concatenation

In the Poincaré model, Shimizu et al. (2021) proposed Poincaré $\beta$-concatenation and $\beta$-split, both of which are numerically unstable in deep networks. As for the Lorentz model, we remark that one could also define operations in the tangent space similarly to the Poincaré $\beta$-concatenation and $\beta$-split. More specifically, if we want to concatenate the input vectors $\{\boldsymbol{x}_i\}_{i=1}^N$ where each $\boldsymbol{x}_i \in \mathbb{L}_K^{n_i}$, we could follow a "Lorentz tangent concatenation": first lift each $\boldsymbol{x}_i$ to the tangent space of the origin $\boldsymbol{o}$: $\boldsymbol{v}_i = \log_{\boldsymbol{o}}^K(\boldsymbol{x}_i) = \begin{bmatrix} v_{i_t} \\ \boldsymbol{v}_{i_s} \end{bmatrix} \in \mathbb{R}^{n_i+1}$, and then perform the Euclidean concatenation to get $\boldsymbol{v} := \left(0, \boldsymbol{v}_{1_s}^\top, \ldots, \boldsymbol{v}_{N_s}^\top\right)^\top$. Finally, we would get $\boldsymbol{y} = \exp_{\boldsymbol{o}}^K(\boldsymbol{v})$ as a concatenated vector in the hyperlolic space. Similarly, we could perform the "Lorentz tangent split" on an input $\boldsymbol{x}_i \in \mathbb{L}_K^n$ with split sub-dimensions $\sum_{i=1}^N n_i = n$ to get $\boldsymbol{v} = \log_{\boldsymbol{o}}^K(\boldsymbol{x}) = \left(0, \boldsymbol{v}_{1_s}^\top \in \mathbb{R}^{n_1}, \ldots, \boldsymbol{v}_{N_s}^\top \in \mathbb{R}^{n_N}\right)^\top$, $\boldsymbol{v}_i = \begin{bmatrix} 0 \\ \boldsymbol{v}_{i_s} \end{bmatrix} \in \mathcal{T}_{\boldsymbol{o}}\mathbb{L}_K^{n_i}$, and the split vectors $\boldsymbol{y}_i = \exp_{\boldsymbol{o}}^K(\boldsymbol{v}_i)$ successively.

Unfortunately, both the Lorentz tangent concatenation and the Lorentz tangent split are not "regularized", which means that the norm of the spatial component will increase after concatenation, and decrease after split. This will make the hidden embeddings numerically unstable. While this problem could be solved by adding regulations, a bigger issue with the Lorentz tangent concatenation and split is that if we use them in a deep neural network, there would be too many exponential and logarithmic maps. Moreover, the tangent space is chosen at $\boldsymbol{o}$. If the points to concatenate are not close to $\boldsymbol{o}$, their hyperbolic relation may not be captured very well. Therefore, we abandon the use of the tangent space and propose more direct and numerically stable operations, which we call the "Lorentz direct concatenation and split" and define as follows: given the input vectors $\{\boldsymbol{x}_i\}_{i=1}^N$ where each $\boldsymbol{x}_i \in \mathbb{L}_K^{n_i}$ and $M = \sum_{i=1}^N n_i$, the Lorentz direct concatenation of $\{\boldsymbol{x}_i\}_{i=1}^N$ is defined to be a vector $\boldsymbol{y} \in \mathbb{L}_k^M$ given by

$$\boldsymbol{y} = \text{HCat}(\{\boldsymbol{x}_i\}_{i=1}^N) = \left[\sqrt{\sum_{i=1}^N x_{i_t}^2 + (N-1)/K}, \boldsymbol{x}_{1_s}^{\text{T}}, \cdots, \boldsymbol{x}_{N_s}^{\text{T}}\right]^{\text{T}}. \tag{1}$$

Note that each $\boldsymbol{x}_{i_s}$ is the spatial component of $\boldsymbol{x}_i$. If we consider $\boldsymbol{x}_i \in \mathbb{L}_K^{n_i}$ as a point in $\mathbb{R}^{n_i+1}$, the projection of $\boldsymbol{x}_i$ onto the Euclidean subspace $\{0\} \times \mathbb{R}^n$, or the closest point there, is $\boldsymbol{x}_{i_s}$. The Lorentz direct concatenation can thus be considered as Euclidean concatenation of the projected points. Lastly, the Euclidean concatenated point is mapped back to $\mathbb{L}_K^M$ by the inverse map of the projection. We remark that this concatenation directly inherits from the Lorentz model, which is one advantage that the Poincaré model does not have.

We also define the Lorentz Split for completeness, though our main focus is on concatenation: Given an input $\boldsymbol{x} \in \mathbb{L}_K^n$, the Lorentz Direct Split of $\boldsymbol{x}$, with sub-dimensions $n_1, \cdots, n_N$ where $\sum_{i=1}^N n_i = n$, will be $\{\boldsymbol{y}_i\}_{i=1}^N$, where each $\boldsymbol{y}_i \in \mathbb{L}_K^{n_i}$ is given by first splitting $\boldsymbol{x}$ in the space dimension as $\boldsymbol{x} = \left[x_t, \boldsymbol{y}_{1_s}^{\text{T}}, \cdots, \boldsymbol{y}_{N_s}^{\text{T}}\right]^{\text{T}}$, and then calculating the corresponding time dimension as $\boldsymbol{y}_i = \begin{bmatrix} \sqrt{\|\boldsymbol{y}_{i_s}\|^2 - 1/K} \\ \boldsymbol{y}_{i_s} \end{bmatrix}$.

## 3. Numerical Stability

We use the following simple experiment to show the advantage of our Lorentz Direct Concatenation over the Lorentz tangent concatenation. The hyperbolic neural network in this simple experiment consists of a cascading of $L$ blocks, and the architecture of each block is as follows: for $l = 0, \cdots, L - 1$,

$$
\begin{aligned}
h_1^{(l)} &= \text{Hlinear}_{d,d}(x^{(l)}), & h_2^{(l)} &= \text{Hlinear}_{d,d}(x^{(l)}); \\
h^{(l)} &= \text{HCat}(h_1^{(l)}, h_2^{(l)}); & x^{(l+1)} &= \text{Hlinear}_{2 \times d, d}(h^{(l)}).
\end{aligned}
\tag{2}
$$

In our test, we take $d = 64$. We sample input and output data from two wrapped normal distributions (Nagano et al., 2019) with different means (input: origin $\boldsymbol{o}$, output: E2H($\mathbf{1}_{64}$)) and variances (input: diag($\mathbf{1}_{64}$), output: $3 \times$ diag($\mathbf{1}_{64}$)). Taking the input as $x^{(0)}$, we fit $x^{(L)}$ to the output data. We record the average gradient norm of the three hyperbolic linear layers in each block. The results for $L = 64$ blocks and $L = 128$ blocks are shown in Figure 1. Clearly, for the first 20 blocks, the Lorentz tangent concatenation leads to significantly larger gradient norms. This difference in norms is clearer when the network is deeper. The gradients from the Lorentz direct concatenation are more stable.

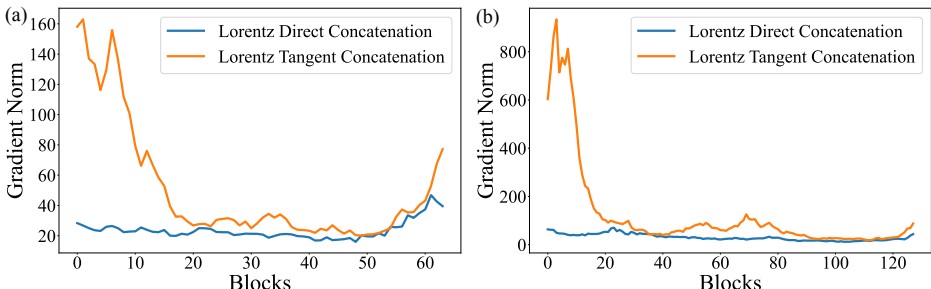

Figure 1: Average gradient norm in training. (a) 64 blocks. (b) 128 blocks.

## 4. Concatenation and Hyperbolic Distances

In this section, we perform additional analysis of Lorentz Direct Concatenation and Lorentz tangent concatenation, particularly their effect on hyperbolic distances.

First, we study the hyperbolic distances to the hyperbolic origin for both concatenation methods. Suppose we have $\boldsymbol{x} \in \mathbb{L}_K^n$ and $\boldsymbol{y} \in \mathbb{L}_K^m$. Let $\boldsymbol{z} = \text{HCat}(\boldsymbol{x}, \boldsymbol{y}) \in \mathbb{L}_K^{n+m-1}$ and $\boldsymbol{z}' = \text{HTCat}(\boldsymbol{x}, \boldsymbol{y}) \in \mathbb{L}_K^{n+m-1}$ be their hyperbolic direct concatenation and hyperbolic tangent concatenation, respectively. We compare the difference between $d_{\mathcal{L}}(\boldsymbol{z}, \boldsymbol{o})$ and $d_{\mathcal{L}}(\boldsymbol{z}', \boldsymbol{o})$ as follows. Note that the distance between an arbitrary point $\boldsymbol{x} \in \mathbb{L}_K^n$ and the origin only depends on the time component:

$$
d_{\mathcal{L}}(\boldsymbol{x}, \boldsymbol{o}) = \frac{1}{\sqrt{-K}} \cosh^{-1}(K \langle \boldsymbol{x}, \boldsymbol{o} \rangle_{\mathcal{L}}) = \frac{1}{\sqrt{-K}} \cosh^{-1}(-K x_t).
\tag{3}
$$

Hence, the distance information is completely contained in the time component. After the concatenation, the time component is $\sqrt{x_t^2 + y_t^2 + 1/K}$. Consequently, for Lorentz Direct

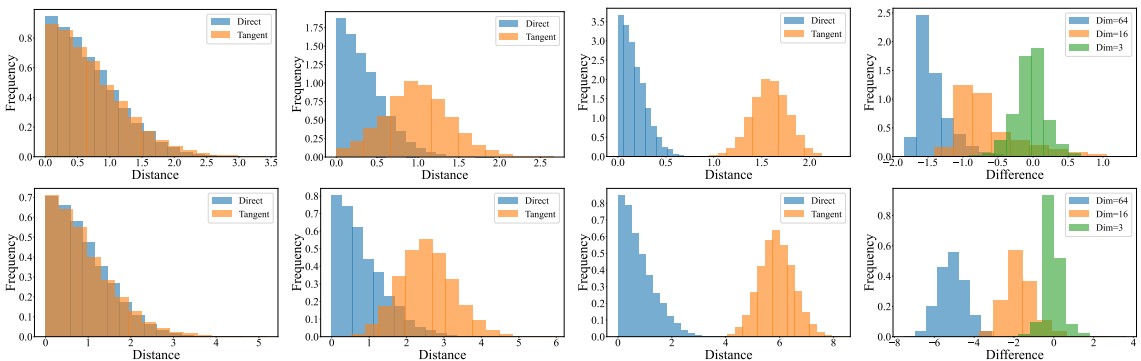

Figure 2: First row: spatial normal. Second row: wrapped normal. Column 1-3: Difference between concatenated distances and original distances with $n = 3, 6, 64$, respectively. Column 4: $|d_{\mathcal{L}}(\boldsymbol{x}_c, \boldsymbol{y}_c) - d_{\mathcal{L}}(\boldsymbol{x}, \boldsymbol{y})| - |d_{\mathcal{L}}(\boldsymbol{x}'_c, \boldsymbol{y}'_c) - d_{\mathcal{L}}(\boldsymbol{x}, \boldsymbol{y})|$.

Concatenation, the distance is

$$d_{\mathcal{L}}(\boldsymbol{z}, \boldsymbol{o}) = \frac{1}{\sqrt{-K}} \cosh^{-1}\left(-K\sqrt{x_t^2 + y_t^2 + 1/K}\right). \tag{4}$$

For Lorentz tangent concatenation, since both the logarithmic and exponential maps reserve the distances, one has

$$\begin{aligned}
d_{\mathcal{L}}(\boldsymbol{z}', \boldsymbol{o}) &= \sqrt{d_{\mathcal{L}}(\boldsymbol{x}, \boldsymbol{o})^2 + d_{\mathcal{L}}(\boldsymbol{y}, \boldsymbol{o})^2} \\
&= \sqrt{\frac{1}{-K}\left(\cosh^{-2}(-Kx_t) + \cosh^{-2}(-Ky_t)\right)}.
\end{aligned} \tag{5}$$

Although the hyperbolic distance $d_{\mathcal{L}}(\boldsymbol{z}, o)$ is not the squared sum of $d_{\mathcal{L}}(\boldsymbol{x}, \boldsymbol{o})$ and $d_{\mathcal{L}}(\boldsymbol{y}, \boldsymbol{o})$, $d_{\mathcal{L}}(\boldsymbol{z}, \boldsymbol{o})$ is larger than each of $d_{\mathcal{L}}(\boldsymbol{x}, \boldsymbol{o})$ and $d_{\mathcal{L}}(\boldsymbol{y}, \boldsymbol{o})$. On the other hand, after concatenation, $d_{\mathcal{L}}^2(\boldsymbol{z}', \boldsymbol{o}) = d_{\mathcal{L}}^2(\boldsymbol{x}, \boldsymbol{o}) + d_{\mathcal{L}}^2(\boldsymbol{y}, \boldsymbol{o})$. This relation agrees with the Euclidean concatenation. However, norm-preservation is not why concatenation works in the Euclidean domain. Therefore, we don't consider this as an advantage of the Lorentz tangent concatenation. The Lorentz Direct Concatenation is more efficient and stable, and no information is lost during concatenation. Therefore, it is still preferred as a neural layer.

More importantly, we study how concatenation changes the relative distances, which is closely related to stability. Specifically, we perform the following experiments. Given $\boldsymbol{x}, \boldsymbol{y}, \boldsymbol{c} \in \mathbb{L}_K^n$, let $\boldsymbol{x}_c = \text{HCat}(\boldsymbol{x}, \boldsymbol{c})$ and $\boldsymbol{y}_c = \text{HCat}(\boldsymbol{y}, \boldsymbol{c})$ be the direct-concatenated version of $(\boldsymbol{x}, \boldsymbol{c})$ and $(\boldsymbol{y}, \boldsymbol{c})$, respectively. Similarly we denote $\boldsymbol{x}'_c = \text{HTCat}(\boldsymbol{x}, \boldsymbol{c})$ and $\boldsymbol{y}'_c = \text{HTCat}(\boldsymbol{y}, \boldsymbol{c})$ for Lorentz tangent concatenations. Since the same vector $\boldsymbol{c}$ is attached to $\boldsymbol{x}$ and $\boldsymbol{y}$, we naturally hope $d_{\mathcal{L}}(\boldsymbol{x}_c, \boldsymbol{y}_c)$ and $d_{\mathcal{L}}(\boldsymbol{x}'_c, \boldsymbol{y}'_c)$ do not deviate much from $d_{\mathcal{L}}(\boldsymbol{x}, \boldsymbol{y})$.

We describe our experiments as follows. Take $K = -1$. We randomly sample three points independently from $\mathbb{L}_K^n$ as $\boldsymbol{x}$, $\boldsymbol{y}$ and $\boldsymbol{c}$ respectively. We have two scenarios for sampling the points: (1) "spatial normal": the points are sampled so that their spatial components follow the standard normal distribution; (2) "wrapped normal": the points

are sampled from the wrapped normal distribution with unit variance. In each scenario, for $n \in \{3, 16, 64\}$, we do the experiments for 10,000 times. We report the distances $|d_{\mathcal{L}}(\boldsymbol{x}_c, \boldsymbol{y}_c) - d_{\mathcal{L}}(\boldsymbol{x}, \boldsymbol{y})|$ and $|d_{\mathcal{L}}(\boldsymbol{x}_c', \boldsymbol{y}_c') - d_{\mathcal{L}}(\boldsymbol{x}, \boldsymbol{y})|$ as well as their differences in Figure 2.

Our experiments clearly show that, especially for large dimensions, the distance between $d_{\mathcal{L}}(\boldsymbol{x}_c, \boldsymbol{y}_c)$ and $d_{\mathcal{L}}(\boldsymbol{x}, \boldsymbol{y})$ is smaller than the distance between $d_{\mathcal{L}}(\boldsymbol{x}_c', \boldsymbol{y}_c')$ and $d_{\mathcal{L}}(\boldsymbol{x}, \boldsymbol{y})$. In particular, in many cases, $|d_{\mathcal{L}}(\boldsymbol{x}_c, \boldsymbol{y}_c) - d_{\mathcal{L}}(\boldsymbol{x}, \boldsymbol{y})|$ is around zero. On the other hand, $|d_{\mathcal{L}}(\boldsymbol{x}_c', \boldsymbol{y}_c') - d_{\mathcal{L}}(\boldsymbol{x}, \boldsymbol{y})|$ tend to be large when $n = 16, 64$, especially when samples follow the wrapped normal distribution. From this result, the Lorentz Direct Concatenation should be preferred to the Lorentz tangent concatenation. In particular, the significant expansion of distance when concatenating with the same vector, in the case $n = 64$, may be one cause of numerical instability.

## 5. Applications of Lorentz Direct Concatenation

In this section, we show that Lorentz direct concatenation can be used in deep and complex neural network architecture with numerical stability. We consider the task of molecular generation using the MOSES dataset (Polykovskiy et al., 2020). Our model contains a tree decoder and a graph decoder are used in the hyperbolic space (the detailed network structure is presented in Appendix B). In both decoders the hyperbolic features need to be concatenated and we consider the following methods for concatenation: $\beta$-concatenation (Shimizu et al., 2021), the Lorentz tangent concatenation and the Lorentz direct concatenation.

The results of molecular generation are presented in Table 1. While Lorentz direct concatenation saturates the important metrics of validity, uniqueness and novelty, the other two concatenation methods both suffer from numerical instability, even if a fully hyperbolic approach is adopted in all the methods.

Table 1: Performance in Validity, Unique(ness), Novelty, SNN. Reported (mean ± std) over three independent samples. "NaN" indicates NaN reported during training.

| Concatenation method | Validity (↑) | Unique (↑) | Novelty (↑) |
|---|---|---|---|
| $\beta$ concat (Shimizu et al., 2021) | NaN | NaN | NaN |
| Lorentz tangent concat | NaN | NaN | NaN |
| **Lorentz direct concat** | 1.0±0.0 | 1.0±0.0 | 0.905±0.006 |

Future work involves more theoretical analysis on the numerical stability of hyperbolic models, as well as models that enjoy both expressivity and stability.

## Acknowledgments

The research results of this article are sponsored by the Kunshan Municipal Government research funding.

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

## Appendix A. Preliminaries of Hyperbolic Geometry

Hyperbolic geometry is a special kind of Riemannian geometry with a constant negative curvature (Cannon et al., 1997; Anderson, 2006). There are five models (coordinate systems) of the hyperbolic space: the Lorentz model, the Poincaré ball model, the Hemisphere model, the Klein model, and the Poincaré half-space model. The models are isometric and the relation between their coordinates is surveyed by Dai et al. (2021). We work with the Lorentz model for its numerical stability.

**The Lorentz Model** The Lorentz model $\mathbb{L}_K^n = (\mathcal{L}, \mathfrak{g}^K)$ of an $n$ dimensional hyperbolic space with constant negative curvature $K$ is an $n$-dimensional manifold $\mathcal{L}$ embedded in the $(n+1)$-dimensional Minkowski space, together with the Riemannian metric tensor $\mathfrak{g}^K = \mathrm{diag}([-1, \mathbf{1}_n^\top])$, where $\mathbf{1}_n$ denotes the $n$-dimensional vector whose entries are all 1's. Every point in $\mathbb{L}_K^n$ is represented by $\boldsymbol{x} = \begin{bmatrix} x_t \\ \boldsymbol{x}_s \end{bmatrix}, x_t > 0, \boldsymbol{x}_s \in \mathbb{R}^n$ and satisfies $\langle \boldsymbol{x}, \boldsymbol{x} \rangle_\mathcal{L} = 1/K$, where $\langle \cdot, \cdot \rangle_\mathcal{L}$ is the Lorentz inner product induced by $\mathfrak{g}^K$:

$$\langle \boldsymbol{x}, \boldsymbol{y} \rangle_\mathcal{L} := \boldsymbol{x}^\top \mathfrak{g}^K \boldsymbol{y} = -x_t y_t + \boldsymbol{x}_s^\top \boldsymbol{y}_s, \ \ \boldsymbol{x}, \boldsymbol{y} \in \mathbb{L}_K^n. \tag{6}$$

**Geodesics and Distances** Geodesics are shortest paths in a manifold, which generalize the notion of "straight lines" in Euclidean geometry. In particular, the length of a geodesic in $\mathbb{L}_K^n$ (the "distance") between $\boldsymbol{x}, \boldsymbol{y} \in \mathbb{L}_K^n$ is given by

$$d_\mathcal{L}(\boldsymbol{x}, \boldsymbol{y}) = \frac{1}{\sqrt{-K}} \cosh^{-1}(K \langle \boldsymbol{x}, \boldsymbol{y} \rangle_\mathcal{L}). \tag{7}$$

**Tangent Space** For each point $\boldsymbol{x} \in \mathbb{L}_K^n$, the tangent space at $\boldsymbol{x}$ is $\mathcal{T}_{\boldsymbol{x}} \mathbb{L}_K^n := \{ \boldsymbol{y} \in \mathbb{R}^{n+1} \mid \langle \boldsymbol{y}, \boldsymbol{x} \rangle_\mathcal{L} = 0 \}$. It is a first order approximation of the hyperbolic manifold around a point $\boldsymbol{x}$ and is a subspace of $\mathbb{R}^{n+1}$. We denote $\|\boldsymbol{v}\|_\mathcal{L} = \sqrt{\langle \boldsymbol{v}, \boldsymbol{v} \rangle_\mathcal{L}}$ as the norm of $\boldsymbol{v} \in \mathcal{T}_{\boldsymbol{x}} \mathbb{L}_K^n$.

**Exponential and Logarithmic Maps** The exponential and logarithmic maps are maps between hyperbolic spaces and their tangent spaces. For $\boldsymbol{x}, \boldsymbol{y} \in \mathbb{L}_K^n$ and $\boldsymbol{v} \in \mathcal{T}_{\boldsymbol{x}} \mathbb{L}_K^n$, the exponential map $\exp_{\boldsymbol{x}}^K(\boldsymbol{v}) : \mathcal{T}_{\boldsymbol{x}} \mathbb{L}_K^n \to \mathbb{L}_K^n$ maps tangent vectors to hyperbolic spaces by assigning $\boldsymbol{v}$ to the point $\exp_{\boldsymbol{x}}^K(\boldsymbol{v}) := \gamma(1)$, where $\gamma$ is the geodesic satisfying $\gamma(0) = \boldsymbol{x}$ and $\gamma'(0) = \boldsymbol{v}$. Specifically,

$$\exp_{\boldsymbol{x}}^K(\boldsymbol{v}) = \cosh(\phi)\boldsymbol{x} + \sinh(\phi)\frac{\boldsymbol{v}}{\phi}, \phi = \sqrt{-K}\|\boldsymbol{v}\|_\mathcal{L}. \tag{8}$$

The logarithmic map $\log_{\boldsymbol{x}}^K(\boldsymbol{y}) : \mathbb{L}_K^n \to \mathcal{T}_{\boldsymbol{x}} \mathbb{L}_K^n$ is the inverse map that satisfies $\log_{\boldsymbol{x}}^K(\exp_{\boldsymbol{x}}^K(\boldsymbol{v})) = \boldsymbol{v}$. Specifically,

$$\log_{\boldsymbol{x}}^K(\boldsymbol{y}) = \frac{\cosh^{-1}(\psi)}{\sqrt{-K}} \frac{\boldsymbol{y} - \psi\boldsymbol{x}}{\|\boldsymbol{y} - \psi\boldsymbol{x}\|_\mathcal{L}}, \psi = K\langle \boldsymbol{x}, \boldsymbol{y} \rangle_\mathcal{L}. \tag{9}$$

## Appendix B. Network Details in Section 5

Our network contains a hyperbolic auto-encoder that learns the hyperbolic embedding of molecules, and a hyperbolic generative network that learns to sample de novo latent embeddings.

### B.1. Details of Hyperbolic Auto-Encoder

**Hyperparameters**
- Manifold curvature: $K = -1.0$
- For all hyperbolic linear layers:
  - Dropout: 0.0
  - Use bias: True
- Optimizer: Riemannian Adam ($\beta_1 = 0.0, \beta_2 = 0.999$)
- Learning rate: 5e-4
- Learning rate scheduler: StepLR (step $= 20000, \gamma = 0.5$)
- Batch size: 32
- Number of epochs: 20

**Graph Encoder**
- Input: graph node features dimension: 35
- Map features to hyperbolic space: $\mathbb{R}^{35} \to \mathbb{L}_K^{35}$
- Hyperbolic GCN layers:
  - Input dimension: 35
  - Hidden dimension: 256
  - Depth: 4
  - Output dimension: 256
- Hyperbolic centroid on all vertices
- Output: graph embedding in $\mathbb{L}_K^{256}$

**Tree Encoder**
- Input: junction tree features dimension 828
- Hyperbolic embedding layer: $\mathbb{R}^{828} \to \mathbb{L}_K^{256}$
- Hyperbolic GCN layers:
  - Input dimension: 256
  - Hidden dimension: 256
  - Depth: 4
  - Output dimension: 256
- Hyperbolic centroid on all vertices
- Output: tree embedding in $\mathbb{L}_K^{256}$

**Tree Decoder**
- Input: tree embedding in $\mathbb{L}_K^{256}$
- Message passing RNN:
  - Input: node feature of current tree node, inward messages
  - Hyperbolic linear layer on inward messages: $\mathbb{L}_K^{256} \to \mathbb{L}_K^{256}$
  - Hyperbolic centroid on inward messages

- – Hyperbolic embedding layer on node feature: $\mathbb{R}^{828} \to \mathbb{L}_K^{256}$
- – Concatenation on node feature and inward message: $\mathbb{L}_K^{256} \to \mathbb{L}_K^{512}$
- – Hyperbolic linear layer: $\mathbb{L}_K^{512} \to \mathbb{L}_K^{256}$
- – Output dimension: 256
- Topological Prediction:
  - – Input: tree embedding, node feature of current tree node, inward messages
  - – Hyperbolic linear layer on inward messages: $\mathbb{L}_K^{256} \to \mathbb{L}_K^{256}$
  - – Hyperbolic centroid on inward messages
  - – Hyperbolic embedding layer on tree feature: $\mathbb{R}^{828} \to \mathbb{L}_K^{256}$
  - – Concatenation on node feature, inward message, and tree embedding: $\mathbb{L}_K^{256} \to \mathbb{L}_K^{768}$
  - – Hyperbolic linear layer: $\mathbb{L}_K^{768} \to \mathbb{L}_K^{256}$
  - – Hyperbolic centroid distance layer: $\mathbb{L}_K^{256} \to \mathbb{R}^2$
  - – Softmax on output
  - – Output dimension: 2
- Label Prediction:
  - – Input: tree embedding, outward messages
  - – Concatenation on outward message, and tree feature: $\mathbb{L}_K^{256} \to \mathbb{L}_K^{512}$
  - – Hyperbolic linear layer: $\mathbb{L}_K^{512} \to \mathbb{L}_K^{256}$
  - – Hyperbolic centroid distance layer: $\mathbb{L}_K^{256} \to \mathbb{R}^{828}$
  - – Softmax on output
  - – Output dimension: 828
- Output: junction tree

**Graph Decoder**
- Input: junction tree, tree message, and graph embedding
- Construction candidate subgraphs
- Hyperbolic graph convolution layers on all subgraphs:
  - – Input dimension: 256
  - – Hidden dimension: 256
  - – Depth: 4
  - – Output dimension: 256
- Hyperbolic centroid on vertices of all subgraphs
- Concatenation on subgraph embedding and graph embedding: $\mathbb{L}_K^{256} \to \mathbb{L}_K^{512}$
- Hyperbolic linear layer: $\mathbb{L}_K^{512} \to \mathbb{L}_K^{256}$
- Hyperbolic centroid distance layer: $\mathbb{L}_K^{256} \to \mathbb{R}$
- Use subgraph score to construct molecular graph
- Output: molecular graph

**B.2. Details of Hyperbolic Generative Adversarial Network**

**Hyperparameters**
- Manifold curvature: $K = -1.0$
- Gradient penalty coefficient: $\lambda = 10$
- For all hyperbolic linear layers:
  - – Dropout: 0.1
  - – Use bias: True

- Optimizer: Riemannian Adam ($\beta_1 = 0, \beta_2 = 0.9$)
- Learning Rate: 1e-4
- Batch size: 64
- Number of epochs: 20
- Gradient penalty $\lambda$: 10

**Generator**
- Input: points sampled from wrapped normal distribution $\mathcal{G}(\boldsymbol{o}, \mathrm{diag}(\mathbf{1}_{128}))$ in $\mathbb{L}_K^{128}$
- Hyperbolic linear layers for graph embedding:
    - Input dimension: 128
    - Hidden dimension: 256
    - Depth: 3
    - Output dimension: 256
- Hyperbolic linear layers for tree embedding:
    - Input dimension: 128
    - Hidden dimension: 256
    - Depth: 3
    - Output dimension: 256
- Output: graph embedding and tree embedding in $\mathbb{L}_K^{128}$

**Discriminator**
- Input: graph embedding and tree embedding in $\mathbb{L}_K^{128}$
- Hyperbolic linear layers for graph embedding:
    - Input dimension: 256
    - Hidden dimension: 256
    - Depth: 2
    - Output dimension: 256
- Hyperbolic linear layers for tree embedding:
    - Input dimension: 256
    - Hidden dimension: 256
    - Depth: 2
    - Output dimension: 256
- Lorentz Direct concatenation on graph embedding and tree embedding: $\mathbb{L}_K^{256} \to \mathbb{L}_K^{512}$
- Hyperbolic linear layer: $\mathbb{L}_K^{512} \to \mathbb{L}_K^{256}$
- Hyperbolic centroid distance layer: $\mathbb{L}_K^{256} \to \mathbb{R}$
- Output: score in $\mathbb{R}$

