# OpenReview forum: "Lorentz Direct Concatenation for Stable Training in Hyperbolic Neural Networks"
_NeurIPS.cc/2022/Workshop/NeurReps — NeurReps 2022 Poster_

### Official Review · Reviewer_Cntv · 2022-10-15
**Making hyperbolic deep networks stable**

**Confidence:** 1
**Soundness:** 3
**Presentation:** 3
**Contribution:** 3
**Overall Rating:** 7

**Summary:**

A new operation is proposed to construct hyperbolic neural networks that improves numerical stability and hence enables the construction of numerically robust deep and complex neural network architectures.

**Questions:**

see above

**Limitations:**

no concern

**Recommended Decision:**

3: Accept

**Relevance:**

3: Solid fit

**Strengths And Weaknesses:**

Disclaimer: I completely misinterpreted the content of the paper when reading the abstract. I am not an expert in this field at all and my report must be taken at best as an educated guess. I apologize for this.

I am quite partial to the questions being discussed here since they are fundamentally about the stability of numerical computations. The single key idea in this abstract appears to be the "Lorentz direct concatenation" in Eq (1). It is a rather obvious operation, but maybe only in hindsight? If this is indeed a new operation then it should of course be studied and this is done in this paper, some general intuition is given why this new ideas leads to improved stability, and a preliminary application. These experiments appear to be carried out carefully and are presented well but I admit that I did not understand in what sense they are related to numerical stability.

E.g., Re Figure 1 - I did not understand how gradient magnitude is related to numerical stability. Variations in the gradient (i.e. the hessian) is related to conditioning. but this is still not the same as numerical stability. Is it possible that the concepts of conditioning and numerical stability are being confused here?


**Submission Track:**

Extended Abstract (4 Page)

---

### Official Review · Reviewer_5SAG · 2022-10-16
**Evaluation**

**Confidence:** 3
**Soundness:** 3
**Presentation:** 2
**Contribution:** 2
**Overall Rating:** 5

**Summary:**

This work mainly studies the instability of concatenation in hyperbolic neural networks. With the identified numerical issue in the previously proposed tangent concatenation in the Poincare model and its direct translation to the Lorenz model, the authors propose direct concatenation with the Lorentz model projection onto subspace $\{0\}\times \mathbb{R}^n$, i.e., the Lorentz direct concatenation. The proposal is intuitive, and the authors have verified its numerical stationarity with an experiment using the MOSES dataset, whereas $\beta$-concatenation in the Poincare model and Lorentz tangent concatenation fail to generate meaningful results.

I think the overall proposal and results seem useful.  If the authors can address the weakness, it shall make an interesting workshop discussion.

**Questions:**

I have discussed in the previous section.

**Limitations:**

Already mentioned in the earlier sections.

**Recommended Decision:**

2: Borderline

**Relevance:**

3: Solid fit

**Strengths And Weaknesses:**

The strengths are already mentioned in the summary section.

While hyperbolic space can intuitively provide an embedding space for a tree-like hierarchical structure, using hyperbolic operations in earlier layers does not seem to have an equally intuitive motivation. Hyperbolic concatenation is even less motivated. Maybe I have missed some critical discussion in the previous literature. I would highly appreciate that if the authors could provide better guidance to the readers and show why hyperbolic operations and concatenation are crucial in building the hyperbolic representation. Further, I highly suggest the authors provide a solid effort to seek a way to falsify the proposal, e.g., to try hard to show why the proposed method is not useful, e.g., with a strong baseline maybe only using hyperbolic representation at the last layer without concatenation. These efforts may further the understanding.

**Submission Track:**

Extended Abstract (4 Page)

---

### Official Review · Reviewer_recx · 2022-10-16
**Lorentz Direct Concatenation for Stable Training in Hyperbolic Neural Networks**

**Confidence:** 5
**Soundness:** 3
**Presentation:** 3
**Contribution:** 3
**Overall Rating:** 7

**Summary:**

In order to extract representation from hierarchical or tree-like data, hyperbolic neural networks are studied. Poincar'e or Lorentz coordinate systems suffer from numerical instability, which makes it challenging to construct hyperbolic neural networks with deep hyperbolic layers. Authors examine the critical process of concatenating hyperbolic representations. As an alternative to concatenating in the tangent space, they suggest the Lorentz direct concatenation and show that it is significantly more stable. they offer some explanations and demonstrate the superiority of carrying out direct concatenation in actual tasks.

**Questions:**

Extend state of the art on Hyperbolic Neural Network by adding references to:
- Equivariant SU(1,1) Neural Network
- Statistics and Machine Learning on SU(1,1) Lie group of Poincaré unit disk

**Limitations:**

Compare with SU(1,1) Equivariant Convolution on Fock-Bargmann Spaces
https://hal.archives-ouvertes.fr/hal-03309817/


**Recommended Decision:**

3: Accept

**Relevance:**

3: Solid fit

**Strengths And Weaknesses:**

Strengths:
Validation on existing data sets:
- the MOSES dataset

Weaknesses:
Comparison should be develeloped with other approaches as
- Equivariant SU(1,1) neural networks:
https://hal.archives-ouvertes.fr/hal-03553274/
https://hal.archives-ouvertes.fr/hal-03309817/
https://link.springer.com/chapter/10.1007/978-3-030-80209-7_62
- Statistics defined in Poincaré Unit Disk:
https://www.sciencedirect.com/science/article/abs/pii/S0169716122000062
https://link.springer.com/chapter/10.1007/978-3-030-80209-7_28


**Submission Track:**

Extended Abstract (4 Page)

---

### Decision · Program_Chairs · 2022-10-21

Accept (Poster)